# Investigation of Measurement Accuracy of Bridge Deformation Using UAV-Based Oblique Photography Technique

**DOI:** 10.3390/s22186822

**Published:** 2022-09-09

**Authors:** Shaohua He, Xiaochun Guo, Jianyan He, Bo Guo, Cheng Zheng

**Affiliations:** 1School of Civil and Transportation Engineering, Guangdong University of Technology, Guangzhou 510006, China; 2Guangdong Polytechnic of Environmental Protection Engineering, Foshan 528216, China

**Keywords:** unmanned aerial vehicle-based oblique photography, 3D reconstruction model, measurement accuracy, bridge deformation

## Abstract

This paper investigates the measurement accuracy of unmanned aerial vehicle-based oblique photography (UAVOP) in bridge deformation identifications. A simply supported concrete beam model was selected and measured using the UAVOP technique. The influences of several parameters, such as overall flight altitude (***h***), local shooting distance (***d***), partial image overlap (***λ***), and arrangement of control points, on the quality of the reconstructed three-dimensional (3D) beam model, were presented and discussed. Experimental results indicated that the quality of the reconstructed 3D model was significantly improved by the fusion overall-partial flight routes (FR), of which the reconstructed model quality was 46.7% higher than those with the single flight route (SR). Despite the minimal impact of overall flight altitude, the reconstructed model quality prominently varied with the local shooting distance, partial image overlap, and control points arrangement. As the ***d*** decreased from 12 m to 8 m, the model quality was improved by 48.2%, and an improvement of 42.5% was also achieved by increasing the ***λ*** from 70% to 80%. The reconstructed model quality of UAVOP with the global-plane control points was 78.4% and 38.4%, respectively, higher than those with the linear and regional control points. Furthermore, an optimized scheme of UAVOP with control points in global-plane arrangement and FR (***h*** = 50 m, ***d*** = 8 m, and ***λ*** = 80%) was recommended. A comparison between the results measured by the UAVOP and the total station showed maximum identification errors of 1.3 mm. The study’s outcomes are expected to serve as potential references for future applications of UAVOP in bridge measurements.

## 1. Introduction

Deformation measurement is critical for newly constructed structures and health monitoring of existing bridges. Generally, the bridge deformation is manually measured using a total station, leveling instrument, displacement sensor, and other measurement equipment. Due to less automation, many inevitable drawbacks such as data gathering limitations, low efficiency, and tedious laboriousness are found for the conventional measuring methods [1,2,3]. Recently, the unmanned aerial vehicle-based oblique photography (UAVOP) technique, identifying structure deformations by reconstructing three-dimensional (3D) models for the objects and comparing the reconstructed models before and after deforming, has gained more and more popularity in engineering practices. The UAVOP technique can better display the object’s overall morphology and local details and possesses high monitoring efficiency, flexible operation, less resource (e.g., time and labor) consumption, and wide application prospects in deformation monitoring [4]. With such merits, the UAVOP technique has gained significant competitive advantages in large regional landforms measurement, of which the monitoring accuracy is within a centimeter [4,5]. Unfortunately, the centimeter-level accuracy of the UAVOP technique cannot satisfy the millimeter-level requirement of bridge monitoring. It is critical to improve the accuracy of UAVOP before applying the technique to bridge measurement [6].

Previous studies indicated that the measurement precision of UAVOP is controlled by the reconstruction quality of the 3D model for objects, which is primarily related to the control point arrangement and UAV flight scheme [7,8,9,10]. Because of this, researchers have tried to improve the quality of the reconstructed model by optimizing the arrangement of control points and UAV flight schemes. Oniga et al. [11] assessed the effect of the control point number on the recovered quality of the 3D model. The experimental results indicated that, as the point number increased from 4 to 20, the model quality recovered based on the UAVOP technique increased by 52%. Tonkin and Midgley [12] examined the quality of the recovered model obtained from UAVOP with different control point arrangements. It was reported that a favorable model was achieved using uniformly distributed control points. To evaluate the influence of flight schemes on the 3D model quality, Chen et al. [13] and Rupnik et al. [14] performed a series of experimental studies varying UAV flight altitude and image overlaps. Results indicated that decreasing the flying height of the single flight route and/or increasing image overlap significantly improved the 3D model. Recently, Morgenthal et al. [15] have successfully reconstructed a millimeter-level refined 3D model by introducing a designated encircling flight route. The effectiveness of using fusion overall-partial flight routes to enhance the partial integrity and recovering quality of UAVOP was verified by Zai [16].

So far, many researchers have extensively addressed the quality of reconstructed 3D models, considering individual characteristics and specific measurement precisions of a monitored object [17,18,19]. However, due to the specified measurement requirements of projects in engineering practices, general approaches to instructing the layout of control points and flight planning are currently unavailable. To measure bridges’ line shape, the structural elements’ local deformation must be monitored. Reconstructing a refined 3D model for the entire bridge would cause serious problems, such as large aerial photography work and consumption of computing resources [20]. Reconstructing a refined 3D model for the local region of the target bridge may avoid these problems, and the quality of the local model can be effectively improved by using fusion overall-partial flight routes and optimizing control points arrangement [16,21]. However, owing to the complicated influence of the aerial environment and the high monitoring accuracy requirements, the applicability of fusion overall-partial flight routes and optimized control points arrangement to increase the model quality needs to be verified. Practical approaches to improving the accuracy of the UAVOP technique in bridge measurement remain unavailable.

To this end, this paper investigates the measurement accuracy of bridge deformation identifications using the UAVOP technique by conducting a series of experimental tests on a simply supported concrete beam model. Test parameters were involved in the experimental program, including the arrangement of control points, overall flight altitude, local shooting distance, and partial image overlap. A bunch of images were obtained for the beam, and 3D models for the bridge were reconstructed using the acquired images. The experimental results clarified critical parameters affecting the quality of the reconstructed local 3D model, and optimized arrangements for control points and UAV flight paths were proposed. The proposed scheme’s capability was further examined using a two-stage bridge deformation measurement test.

## 2. Aerotriangulation Method Based on Oblique Photography

Figure 1 shows the fundamental process of bridge measurement using the UAVOP technique. As can be seen, the act of measuring bridge line shapes consists of the following procedures: (a) image acquisition; (b) corresponding points matching; (c) aerotriangulation; (d) 3D model reconstruction; and (e) spatial data comparison. Noticeably, aerotriangulation is the most necessary procedure for successfully reconstructing the 3D bridge models [22].

Theoretically, the aerotriangulation can be resolved using bundle block adjustment, involving a collinear equation as the adjustment base and photography bundle and control point as the constraint. The outcome of aerotriangulation includes elements of exterior orientation and coordinates of target points, with which the 3D sparse point cloud model of the monitored object can be reconstructed.

In the process of solving the aerotriangulation, the bundle block adjustment collinear equations can be described as the following expressions:(1)Φ(T)=L
(2)$$\Phi \left(T\right)=\left[\begin{array}{l} -f\frac{a_{1}\left(X-X_{s}\right)+b_{1}\left(Y-Y_{s}\right)+c_{1}\left(Z-Z_{s}\right)}{a_{3}\left(X-X_{s}\right)+b_{3}\left(Y-Y_{s}\right)+c_{3}\left(Z-Z_{s}\right)}\\ -f\frac{a_{2}\left(X-X_{s}\right)+b_{2}\left(Y-Y_{s}\right)+c_{2}\left(Z-Z_{s}\right)}{a_{3}\left(X-X_{s}\right)+b_{3}\left(Y-Y_{s}\right)+c_{3}\left(Z-Z_{s}\right)} \end{array}\right];\;L=\left[\begin{array}{l} x\\ y \end{array}\right]$$
where *x* and *y* are the image point coordinates in lateral and vertical directions, respectively; *f* is the focal length of the camera; *X_s_*, *Y_s_*, *Z_s_*, *X*, *Y,* and *Z* represent the spatial coordinates of the photography center and target point in the object coordinate system; *a_i_*, *b_i_*, and *c_i_* are directional cosines composing of elements of exterior orientation (*φ*, *ω*, *κ*); *T* denotes the adjustment elements including *X*, *Y*, *Z*, *f*, *X_s_*, *Y_s_*, *Z_s_*, *φ*, *ω*, and *κ*.

Since the projection position of target points always deviates from the ideal position caused by the image noise [23], the modification value of *L* is now used to modify the error. Accordingly, Equation (1) can be linearized by using the first order Taylor expansion:(3)$$\Phi \left(T\right)=\Phi \left(T^{0}\right)+B\delta_{T}=L+\delta_{L}$$
where *T*^0^ is the initial value of the adjustment elements; *δ_T_* is the modification value of *T*; *δ_L_* is the modification value of *L*; $B=\frac{\partial \Phi }{\partial T}{|}_{T^{0}}$.

With the help of the least squares method, the modification value of *T* can be acquired by minimizing *δ_L_*, and the calculation process is re-performed using the new initial value (*T*^1^). After constant iteration, the elements of exterior orientation and coordinates of target points can be obtained when *δ_T_* is smaller than the given threshold.

The error between the reprojected and the ideal position of the target points, normally described as reprojection error, represents the precision of aerotriangulation. As the spatial resolutions vary with the flight plan used for UAVOP, the same reprojection error of target points may produce different aerotriangulation precision for the monitored object. As shown in Figure 2, a stereo vision system consisting of two cameras is used to visualize the relationship between reprojection error and aerotriangulation precision.

As shown in Figure 2, A is the target point to be solved; *a_L*0*_* and *a_R*0*_* are the ideal positions of A point in left and right images, respectively; ‖*AA_L_*‖ is the actual deviation on the surface of the monitored object, corresponding to the distance from *a_L_* to *a_L*0*_*; ‖*AA_R_*‖ is the actual deviation on the surface of the monitored object, corresponding to the distance from *a_R_* to *a_R*0*_*. The actual deviation on the object surface directly reflects the precision of aerotriangulation, and less deviation means better aerotriangulation precision.

The actual deviation of the stereo vision system can be expressed as:(4)$$A_{e}=\left\|AA_{L}\right\|+\left\|AA_{R}\right\|$$
(5)$$\left\|AA_{L}\right\|\approx d_{L}k_{L}\omega_{L}$$
(6)$$\left\|AA_{R}\right\|\approx d_{R}k_{R}\omega_{R}$$
(7)$$\omega_{L}=\frac{\sin(2\tan^{-1}\frac{l}{2f})}{l\cos(\tan^{-1}\frac{l}{2f}-\eta_{L})\cos(\tan^{-1}\frac{l}{2f}+\eta_{L})}$$
(8)$$\omega_{R}=\frac{\sin(2\tan^{-1}\frac{l}{2f})}{l\cos(\tan^{-1}\frac{l}{2f}-\eta_{R})\cos(\tan^{-1}\frac{l}{2f}+\eta_{R})}$$
where *A_e_* is the actual deviation of the stereo vision system on the object surface; *d_L_* and *d_R_* are the distance from the location of image collecting to the object surface; *k_L_* and *k_R_* are the reprojection error; *η_L_* and *η_R_* are the tilt angle of image; *l* is the size of the image; *f* is the focal length of the camera.

Combining Equations (4)–(8), it is easily found that the actual deviation is controlled by the shooting distance, reprojection error, and the tilt angle of the image. According to the principle of the least square method, there is a threshold for optimizing the reprojection error that is determined by a specified confidence level. The experimental results reported by Barba et al. [24] showed that using threshold *k* = 0.46 pixels produced a 95% confidence level for the measurement. The image angle can be fixed in a mission of aerial photography, and the tilt angle of the image usually ranges from 4.5° to 49.5° in practice [25]. To examine the influence of the shooting distance, images with a tilt angle value of 15°, 30°, 45°, *f* = 35 mm, *l* = 35.5 mm, and *k* = 0.46 pixels were substituted into the above equations, and the achieved theoretical results are presented in Figure 3.

From Figure 3, one can see that the actual deviation of the stereo vision system is proportional to the shooting distance under fixed image angles, indicating that the precision of aerotriangulation decreased with the enlargement of the distance between the UAV and the monitored object. Meanwhile, more control points and higher image overlap produce higher aerotriangulation precision, which contributes to different reconstruction models. To quantify the impact of parameters on the quality of reconstructed 3D models, the arrangement of control points, flight route altitude, the distance between the camera and monitored object, and the image overlap are selected and evaluated in the current experimental program.

## 3. Experimental Work

### 3.1. Monitored Object and Instrumentations

An experimental test to investigate the influence of critical parameters on the reconstructed 3D model quality was conducted using the UAVOP. The monitoring object and instrumentation of the experimental program are shown in Figure 4. A simply-supported reinforced concrete beam was fabricated and selected as the monitored object. The length × width × depth of the concrete beam was 1500 mm × 100 mm × 200 mm. The selected beam was positioned in a yard with a length of 15 m and a width of 8 m, to minimize the disturbance of stuff around. Laminated rubber bearings with a dimension of 150 mm × 200 mm × 28 mm were used at beam support, and square markers on the beam’s side surface were used as control points and checkpoints. The marker size needs to consider the local shooting distance, which influences the scale of the control point in an image [26]. Accordingly, the size of markers was determined as 50 mm × 50 mm, considering the shooting distance and the size of the concrete beam model.

The actual coordinates of control points and checkpoints were measured using the Southern NTS-342R total station without a prism (see Figure 4b), which has a precision of 3.0 mm ± 2.0 ppm. Aerial images of the beam were taken by the DJI Zenmuse P1 camera equipped on the DJI M300-RTK UAV (see Figure 4c). The focal length and effective pixels of the P1 camera are 35 mm and 45 million, respectively. The real-time kinematic (RTK) module of the DJI M300-RTK UAV can receive navigation satellite signals, which possess high precision in locating the UAV during the operation. The three-axis stabilized gimbal of the P1 camera can adjust the shooting angle from negative 130 to positive 40 degrees in the pitch range, while that in the cross-roll range is from negative 55 to positive 55 degrees for the camera. The flight route plan and UAV control of the test were realized using the DJI Pilot App flight control software [27].

### 3.2. Test Parameters

#### 3.2.1. UAV Flight Route Planning

A summary of testing parameters for the current experimental program is shown in Table 1. To quantify the usefulness of the fusion flight route (FR) in improving the reconstructed model quality, images of the monitored object were taken by the UAVOP with a single flight route (SR) and FR. The FR collected images from both the overall and partial flight routes, while the SR collected images only from the overall flight route. According to the flight requirements of the UAV instrument, the minimum flying altitude should be larger than 1.5 times the maximum height of structures within the flight area [13]. For the current test, the overall flight altitude (***h***) of the overall flight route was selected as 10 m, 30 m, and 50 m, respectively, which can satisfy the safe flying altitude of the flight area. The image overlaps of the overall flight route for forward and sidelap were set as 90% and 80%, respectively. To achieve clear images for the 50 mm × 50 mm markers, the maximum distance from the camera to the targets must be smaller than 10 m. Accordingly, the local shooting distance (***d***) for the partial flight route was 4 m, 8 m, and 12 m, respectively. Moreover, previous studies indicated that the overlapping region of the contiguous image should be larger than 60% [14]. In this work, the image overlap of the partial flight route (***λ***) was designated as 70%, 80%, and 90%, respectively. Figure 5 shows the schematic diagram of testing parameters.

#### 3.2.2. Arrangement of Reference Points

Reference points include control points on the side surface of the monitored beam and checkpoints on surfaces of the concrete blocks and beam (see Figure 4a and Figure 6). The control points aim to improve the precision of aerotriangulation, and the checkpoints distributed over the entire field aim to evaluate the overall and partial quality of the 3D model. Figure 6 and Table 2 show the detailed arrangement of reference points. The coordinates for all reference points were located using the total station before launching the UAV.

Due to the uncertain influence of complicated environments and traffic vehicles, it is impractical to keep control points on the top surface of bridge decks for a comparatively long monitoring period. The control points on the bridge’s side surfaces are easy to protect during the bridge’s construction and operation stages. Consequently, as shown in Table 2, the control points were attached to the side surface of the concrete beam. A total of fourteen reference points were used. Specifically, the location of control points is categorized as global layout (G), regional layout (R), linear layout (L), and planar layout (P). The control points include 3, 6, and 8 for different point arrangements.

ASPRS standard is widely adopted in the assessment of UAV photogrammetry studies [28,29]. The standard employs the root mean square error (*RMSE*) for the checkpoints coordinates to check the quality of the reconstructed model. Formulas to determine the *RMSE* are described as:(9)$$RMSE_{xy}=\sqrt{\frac{\sum_{i=1}^{n}\Delta x_{i}^{2}+\sum_{i=1}^{n}\Delta y_{i}^{2}}{n}}$$
(10)$$RMSE_{z}=\sqrt{\frac{\sum_{i=1}^{n}\Delta z_{i}^{2}}{n}}$$
(11)$$RMSE_{xyz}=\sqrt{\frac{\sum_{i=1}^{n}\Delta x_{i}^{2}+\sum_{i=1}^{n}\Delta y_{i}^{2}+\sum_{i=1}^{n}\Delta z_{i}^{2}}{n}}$$
where *n* is the number of checkpoints; ∆*x_i_*, ∆*y_i_*, ∆*z_i_* are the three-dimensional coordinate differences of the *i*th checkpoint between the reconstructed model and practical value in directions of *X*, *Y*, and *Z*, respectively.

### 3.3. Discussions on the Reconstructed 3D Model Based on UAVOP

#### 3.3.1. Effect of Fusing Overall-Partial Flight Route

Figure 7 shows the 3D models recovered based on the images obtained by UAVOP with FR (***h*** = 50 m, ***d*** = 4 m, and ***λ*** = 80%) and SR (***h*** = 50 m). As can be seen, the 3D model reconstructed using data obtained from the FR exhibits favorable integrity and clear facade texture. In contrast, the model recovered based on the SR route is blurry and meaningless. Table 3 summarizes the quality of the reconstructed 3D model from UAVOP with different overall flight altitudes. Under the identical overall flight altitude, the average reconstruction quality of the 3D beam model achieved from the FR is 46.7% higher than those using the SR. Adopting the FR in UAVOP effectively improved the reconstructed model quality. Increasing the flight altitude from 10 m to 50 m, the model recovered from the FR remained stable roughly. In contrast, the quality of the model from the SR decreased by 53.7%. A possible explanation may be that, as the distance between the UAV and the monitored object became large, the object’s detailed features and texture variations failed to be captured, resulting in the low model quality of the SR. The partial route of FR provided more identifiable details in processing the model reconstruction, which was barely affected by the flight altitude of the overall flight route. The experimental results conclude that the UAVOP with FR is more suitable for measuring the bridge deformation with limited flight altitude by the site.

#### 3.3.2. Effect of Local Shooting Distance

The *RMSE* of the 3D models with different local shooting distances and partial image overlap was used to evaluate the model quality. The influence of the partial flight route on the reconstructed model quality is presented in Figure 8. As can be seen, the model quality is improved as a short local shooting distance (***d***) is used. By decreasing the local shooting distance from 12 m to 8 m, the model quality increased by 48.2%, and the further decrease of local shooting distance from 8 m to 4 m brought a few changes to the model. A possible explanation might be that using a shorter local shooting distance resulted in more pixels for the concrete beam in captured images. The further increase of pixels contributed to a higher reconstructed beam model quality. It is worth noting that a critical ***d*** restricts the upper limit quality of the reconstructed model, making it impractical to continuously improve the model by decreasing the shooting distance after the value of ***d*** is below the threshold. For a bridge deformation measurement using 50 mm × 50 mm markers, the reasonable local shooting distance is suggested as 8 m.

#### 3.3.3. Effect of Partial Image Overlap

As shown in Figure 8, the quality of the reconstructed 3D model is improved as higher partial image overlap (***λ***) is used. Under a fixed shooting distance of ***d*** = 8 m, increasing the ***λ*** from 70% to 80% improved the model quality by 42.5%, and the model was further enhanced by 3.3% when the ***λ*** continually increased to 90%. Increasing image overlap can improve the number of corresponding points between multi-view images. The gradually increased corresponding points improved the quality of image matching in the aerotriangulation, which contributed to the good quality of the reconstructed 3D model. From the perspective of reducing the computational works of model reconstructions, the partial route of ***d*** = 8 m and ***λ*** = 80% is recommended for the UAVOP with the FR route.

#### 3.3.4. Effect of Control Point Arrangement

The influence of control points arrangement is presented in Figure 9. By comparing the reconstructed quality of the overall field and concrete beam, it is easily found that the identified coordinates of the checkpoints adjacent to the control points matched well with their actual locations. Arranging control points on the concrete beam can improve the quality of the reconstructed 3D model. Similar to the results reported by many other researchers [7,9,11], the 3D model gradually improved as the number of control points increased. Specifically, increasing the number of control points from 3 to 6 improved the model quality by 19.4%, and a further increase of points from 6 to 8 improved the model quality by 8.0%. The GL3 model had the lowest recovery quality may be explained by the poorly resolved coordinates resulting from the anomalous aerotriangulation coordinate system. Compared to the GL3 model, the quality of the GP3 model was improved by 78.4%. It is significant to use planar arranged control points to achieve a good coordinate system and improve the quality of the reconstructed model. The concrete beam with control points in a global arrangement showed a 38.4% higher quality than those with the regional points. Hence, the global-plane control points are necessary to achieve high reconstructed model quality.

## 4. Bridge Deformation Measurement Using a UAVOP Technique

The optimizing scheme with the GP3 control points arrangement and the FR (***h*** = 50 m, ***d*** = 8 m, and ***λ*** = 80%) was used to examine the applicability in bridge deformation measurements. The test verification consisted of two stages: (i) *Stage I* for the beam supported by two bridge rubber supports, and (ii) *Stage II* for the beam with one rubber support removed to generate deformations. The two-stage images of the concrete beam were obtained using the optimizing scheme, which provided the necessary data for reconstructing the 3D beam models. The deformation was identified by comparing the two-stage reconstructed models. For the convenience of comparison, the scheme with GP3 control points arrangement and the SR (***h*** = 50 m) was adopted as a counterpart. The quality of the reconstructed 3D models obtained from the test is summarized in Table 4.

As can be seen in Table 4, the difference between the *RMSE_xyz_* obtained from *Stages I* and *II* in FR models is 0.04 cm, and the corresponding difference for the SR model is 0.22 cm. The comparatively close *RMSE_xyz_* of models before and after deforming demonstrated that the working conditions of the two stages are similar, and the quality of the reconstructed models is reliable. To analyze the deformation of the reconstructed beam model, as shown in Figure 10a, the top contour line obtained from the two-stage model is used to represent the concrete beam. One can see that the identified beam shape after the deformation of the FR model correlated well with the actual shape of the beam. In contrast, the SR model produced quite different results after deformation occurred to the beam. The different outcomes of the FR and SR models reveal that the accuracy of deformation measurement is significantly affected by the model quality. Figure 10b compares the reconstructed models of two stages based on the optimized FR. The concrete beam’s overall deformation is more intuitive than the single-point measurement with a total station.

The data obtained from the total station instrument were defined as the baseline and provided references for the 3D model. Errors of the elevation variation between the reconstructed 3D beam model and the actual value are calculated and presented in Figure 11. As can be seen, the variation errors at each point fail to correlate well with the errors of the contour line shape. An explanation is the uneven model quality and the point errors of manual selection. The maximum errors of deformation obtained from the SR model and FR model are 13 mm and 1.3 mm, respectively. The deformation measurement accuracy of the SR model reaches a centimeter, while that of the FR model approaches the millimeter, which meets the accuracy requirements of bridge deformation measurement.

## 5. Conclusions

This paper investigates the accuracy of bridge deformation measurement by a UAV-based oblique photography (UAVOP) technique by conducting a model test on a simply supported reinforced concrete beam. Testing parameters varying overall flight altitude (***h***), local shooting distance (***d***), partial image overlap (***λ***), and the arrangement of the control points were involved in the experimental program. An optimized scheme that included UAV flight paths and arrangements for control points was proposed, and the scheme’s applicability in deformation measurement was verified. The conclusions are drawn as follows:

(1) The reconstructed 3D model quality was significantly improved using UAVOP with fusion overall-partial flight routes (FR).

(2) The reconstructed model quality was significantly affected by the local shooting distance, partial image overlap, and the arrangement of control points, while the impact of overall flight altitude was small. Decreasing the shooting distance from 12 m to 8 m, the quality of the reconstructed 3D model was improved by 48.2%, and the model quality was also increased by 42.5% as the value of ***λ*** increased from 70% to 80%. Compared to the models with linearly or regionally arranged control points, the quality of models with global-plane control points improved by 78.4% and 38.4%, respectively.

(3) Overall-partial flight routes and control points arranged on the bridge facade in a global-plane are needed to improve the quality of the reconstructed model. The recommended optimized scheme for bridge measurement is control points in GP3 arrangement, and the FR (***h*** = 50 m, ***d*** = 8 m, and ***λ*** = 80%).

(4) The accuracy of deformation measurement was significantly affected by the quality of the reconstructed model, and the maximum error of deformation identification obtained from the optimized model was 1.3 mm.

This study evaluates the feasibility of using UAVOP to measure bridge deformations. The effectiveness of using the FR and global-plane control points arrangement in improving the measurement accuracy was confirmed. The achievements of this study provide a potential reference for the future application of UAVOP in bridge deformation measurements. In future studies, more field tests on actual bridges are suggested to verify the conclusions of the present work.

## Figures and Tables

**Figure 1 sensors-22-06822-f001:**
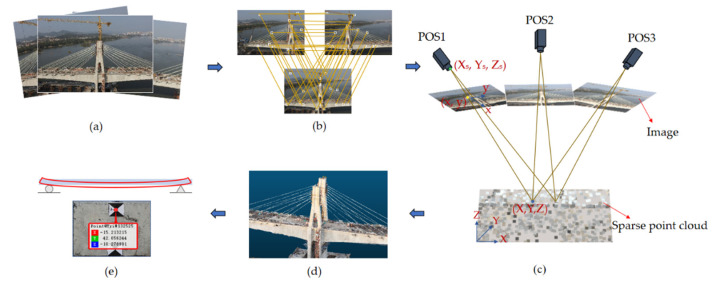
The process of deformation measurement based on oblique photography: (**a**) multi-view image acquisition; (**b**) corresponding points matching; (**c**) aerotriangulation; (**d**) 3D model reconstruction; (**e**) spatial data comparison.

**Figure 2 sensors-22-06822-f002:**
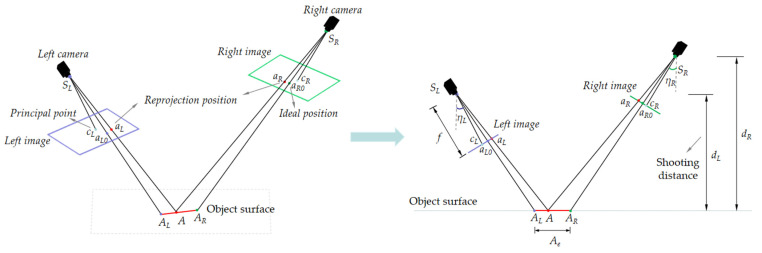
Aerotriangulation of the stereo vision system in space (**left**) and plane (**right**).

**Figure 3 sensors-22-06822-f003:**
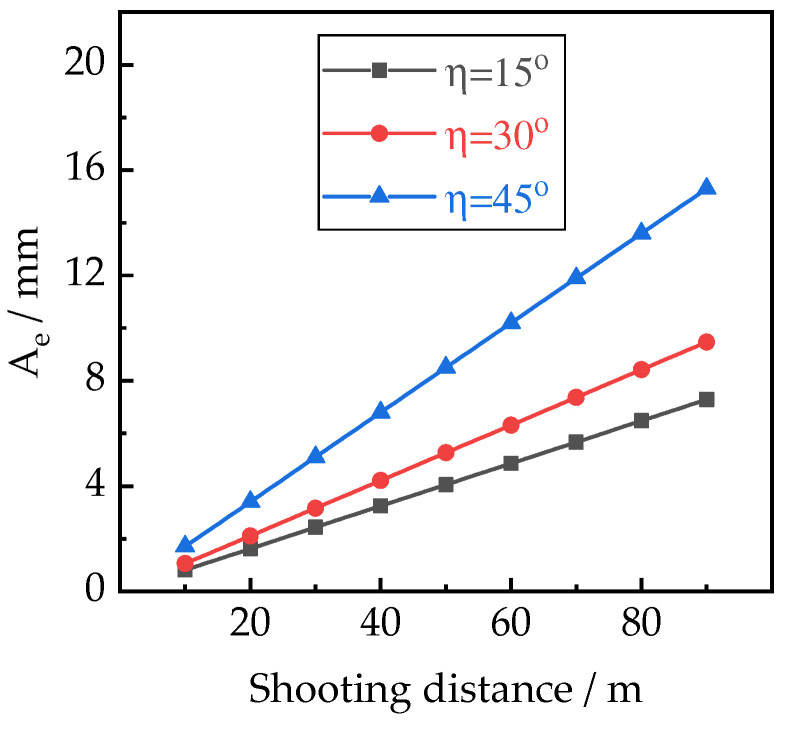
The effect of shooting distance.

**Figure 4 sensors-22-06822-f004:**
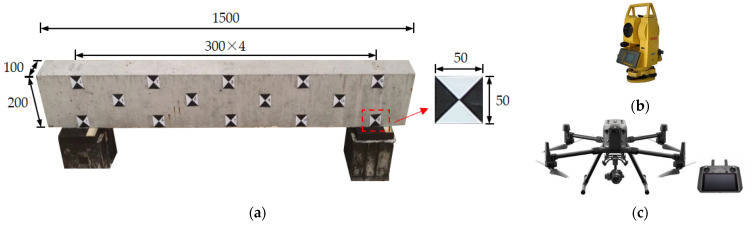
Test object and instrumentations: (**a**) simply-supported concrete beam (Unit: mm); (**b**) total Station Instrument; (**c**) UAV aerial photography system.

**Figure 5 sensors-22-06822-f005:**
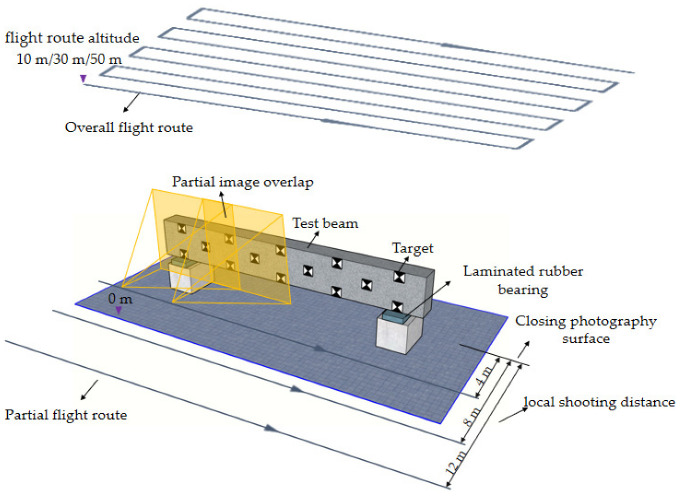
Schematic diagram of testing parameters.

**Figure 6 sensors-22-06822-f006:**
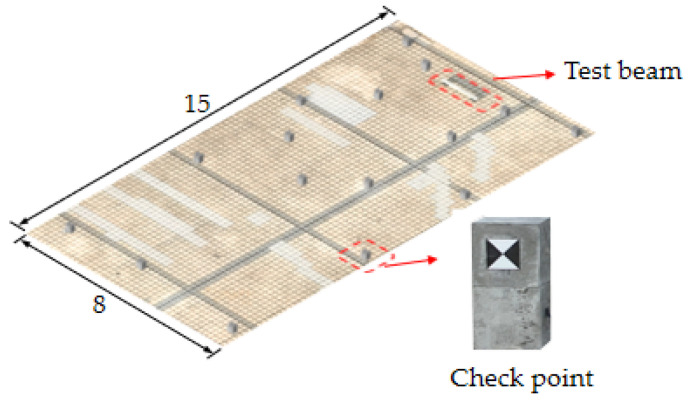
The detailed arrangement of the checkpoints (Unit: m).

**Figure 7 sensors-22-06822-f007:**
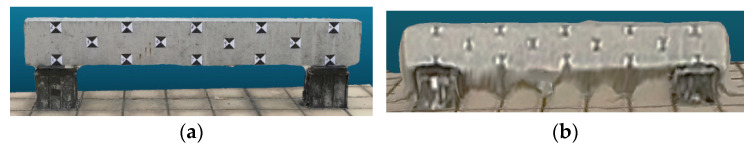
3D model of concrete beam: (**a**) the model of FR; (**b**) the model of SR.

**Figure 8 sensors-22-06822-f008:**
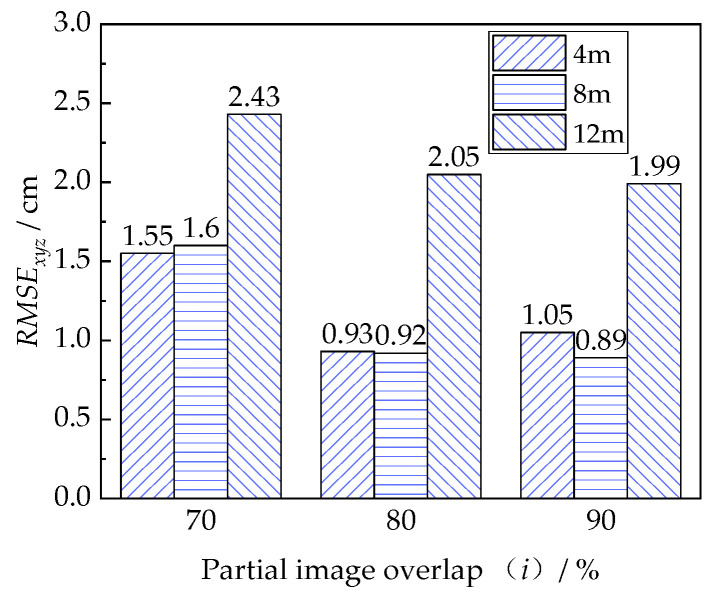
The effect of the partial route on the quality of the 3D beam model.

**Figure 9 sensors-22-06822-f009:**
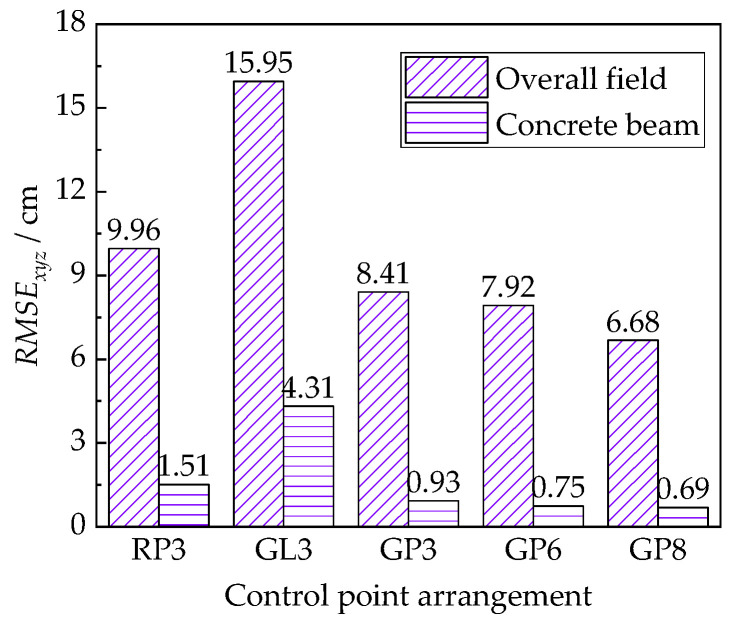
The effect of control point arrangement on 3D model quality.

**Figure 10 sensors-22-06822-f010:**
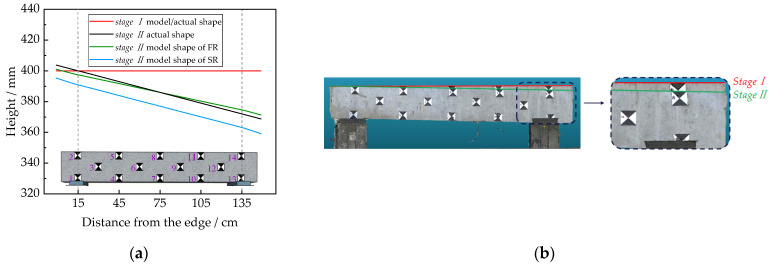
Deformation measurement: (**a**) deformation of the two-stage model; (**b**) comparison of the two-stages models of FR.

**Figure 11 sensors-22-06822-f011:**
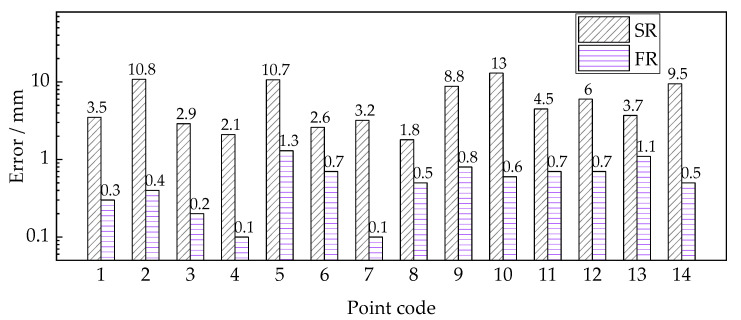
Errors of points deformation measurement by using the reconstructed 3D beam models.

**Table 1 sensors-22-06822-t001:** Summary of parameters for experiments.

Parameter	Number of Control Points	Layout of Control Points	Overall Flight Altitude (*h*)/m	Partial Flight Route	
				Local Shooting Distance (*d*)/m	Partial Image Overlap (*λ*)
Control point	3, 6, 8	See Table 2	50	4	80%
Overall flight altitude (*h*)	3	GP3	10, 30, 50	4	80%
Local shooting distance (*d*)	3	GP3	50	4, 8, 12	70%, 80%, 90%
Partial image overlap (*λ*)	3	GP3	50	4, 8, 12	70%, 80%, 90%

**Table 2 sensors-22-06822-t002:** Arrangement of reference points on the concrete beam.

Code	Layout Type	Number of Control Points	Distribution Map
RP3	Regional and planar	3	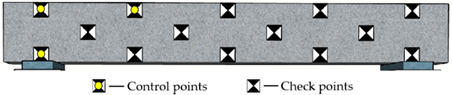
GL3	Global and linear	3	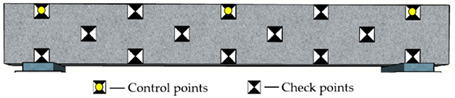
GP3	Global and planar	3	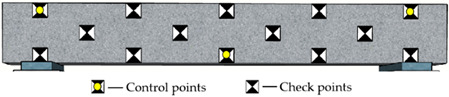
GP6	Global and planar	6	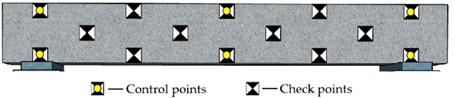
GP8	Global and planar	8	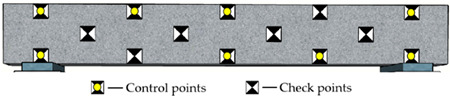

**Table 3 sensors-22-06822-t003:** The effect of different routes on reconstructed quality of the 3D beam model.

*h*/m	*RMSE_xyz_* of FR/cm	*RMSE_xyz_* of SR/cm	The Enhancing Effect of FR/%
10	1.02	1.39	26.6
30	0.90	1.62	44.4
50	0.93	3.00	69.0

**Table 4 sensors-22-06822-t004:** The two-stage model quality of the concrete beam.

	*RMSE_xy_*/cm		*RMSE_z_*/cm		*RMSE_xyz_*/cm	
	FR	SR	FR	SR	FR	SR
*Stage I*	0.75	2.31	0.53	1.99	0.92	3.05
*Stage II*	0.65	2.71	0.59	1.83	0.88	3.27

## Data Availability

Some or all data, models, or code that support the findings of this study are available from the corresponding author upon reasonable request.

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
