# Peer review of "Investigation of Measurement Accuracy of Bridge Deformation Using UAV-Based Oblique Photography Technique"

_sensors, 2022, doi:10.3390/s22186822_

Round 1

Reviewer 1 Report

This work made the investigation of measurement accuracy of bridge deformation using UAV-based oblique photography technique on a simply supported reinforced concrete beam bridge model. The study is interesting and expected to be served as potential references for future application of UAVOP in bridge deformation measurements. I recommend the publication of this work if the authors could properly give some comments on the following problems:

 1)The English expression of the manuscript needs to be improved;

2)The abstract seems too long, so it should be simplified to be more concise.

3) In Section 3.3,  how to appaise the effect of local shooting distance and the effect of control point arrangement ? Is there any evaluation standard?

4) More detailed analysis of the measurement process shold be displayed in Section 4 to examine the applicability of the optimizing scheme.

5)In the conclusions, the innovation of this manuscript should be highlighted, not summarized. In addition, the future research direction should be pointed out.

Reviewer 2 Report

The authors want to improve the measurement accuracy of UAVOP in bridge deformation identifications, and a series of experimental tests are carried out for some key parameters. It is well-written language-wise but some parts need a major clarification before it could be considered for publication.

The reinforced concrete beam model used in experiments is a miniature and the shooting distances are shortened in equal proportion. But the control and check points are not shortened in equal proportion, and I can't imagine a real bridge with so many huge markers on it. Since the control points are relatively large and reduce the difficulty of technical implementation, I doubt whether this work can maintain such accuracy in real bridge deformation measurement.

The data from total station instrument is set as the actual values and the maximum measurement error 3.0 mm. Meanwhile, the authors claim they improve the maximum identification error of bridge displacements to 1.3 mm. Since these two numbers are very close, I doubt whether the total station data can be used as accurate data.

The authors mainly rely on more flight distance to improve the measurement accuracy, the so-called fusing overall-partial flight routes (FR). This is obviously achievable because it consumes a lot more energy. It would be useful to be able to give an optimal flight distance considering measurement accuracy and energy consumption, rather than just talking about flight paths.

Reviewer 3 Report

Dear authors,

I found your manuscript ready for publication at its current form (see only 2 minor comments on the attached pdf file), as both English and Scientific soundness, but also the importance of your results and discussion are well presented. 

Author Response

Dear Reviewer,

We are thankful to the reviewer for providing a helpful appreciation and careful review to our manuscript. As suggested, the numbering has been corrected in the revised manuscript. (Lines 182 and 184)